# TapSampling: Inference-Time Sampling with a Task-Progress-Understanding Verifier for Robotic Manipulation

**Sizhe Zhao**[1]  **Shengping Zhang**[1 2]  **Shuo Yang**[1]  **Weiyu Zhao**[1]  **Shuigen Wang**[3]  **Xiangyang Ji**[4]

## Abstract

Existing embodied control research demonstrates remarkable performance improvements by scaling training data and model size. We instead explore inference-time strategy as an alternative axis. Non-deterministic generative models, such as diffusion and autoregressive models, have been widely adopted in the field of embodied control. However, the single-shot inference paradigm limits their performance. In this paper, we propose **TapSampling**, a plug-and-play framework for inference-time sampling. First, we introduce an Action-VAE that represents actions in a low-dimensional latent space by mapping policy-generated initial actions into a compressed posterior distribution, from which any number of latent samples can be drawn and decoded into candidate actions that approximate the true action distribution. Second, we formulate action verification as task-progress outcome prediction, using the intrinsic sequential structure of robotic datasets to train a semantically grounded verifier for interpretable action selection. Furthermore, TapSampling is a policy-agnostic framework. Extensive experiments in both simulated and real-world environments demonstrate that our method substantially improves multiple generalist policies without further policy finetuning. Code and models are available at the **project page**.

## 1. Introduction

Recently, non-deterministic approaches such as diffusion models and auto-regressive Large Language Models (LLMs)

[1]Harbin Institute of Technology, China [2]Harbin Institute of Technology (Weihai) Qingdao Research Institute, China [3]Iray Technology co., Ltd., Shandong, China [4]Tsinghua University, Beijing, China. Correspondence to: Shengping Zhang <s.zhang@hit.edu.cn>.

*Proceedings of the 43rd International Conference on Machine Learning*, Seoul, South Korea. PMLR 306, 2026. Copyright 2026 by the author(s).

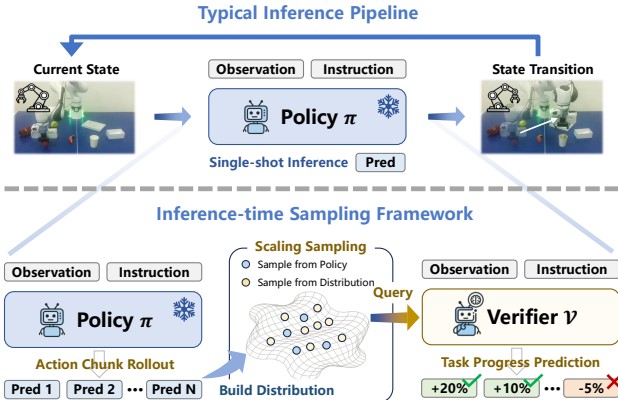

*Figure 1.* **TapSampling**, an inference-time sampling framework, extends the typical single-shot inference pipeline through multi-sample generation and task-progress-guided verification.

have demonstrated remarkable capabilities in perceiving, understanding, and generating multimodal information (Chen et al., 2025; Yin et al., 2024). Capitalizing on these remarkable advancements, there is a burgeoning initiative to harness pretrained foundation models, such as Video Diffusion Models (VDMs) (Blattmann et al., 2023; Yang et al., 2025b) and Vision-Language Models (VLMs) (Karamcheti et al., 2024; Beyer et al., 2024; Bai et al., 2025), as the backbone, and adapt them for robotic manipulation by expanding action vocabularies or incorporating additional action heads (Kim et al., 2024; Black et al., 2025b; Wang et al., 2026; Lin et al., 2026). Driven by the accumulation of large-scale data and advances in model architectures, policy models have demonstrated remarkable capabilities (O'Neill et al., 2024; Kareer et al., 2025). Nevertheless, many policies exhibit considerable instability due to the non-deterministic characteristic of diffusion and next-token-prediction paradigms as shown in the Appendix A, succeeding in some cases and failing in others under identical environment conditions. Therefore, their performance is fundamentally limited due to the single-shot inference paradigm.

Due to the stochastic characteristics of diffusion and next-token-prediction paradigms, the inference-time strategy has emerged as a prominent research focus. Experiments indicate that sampling repeatedly from LLMs and diffusion models effectively improves the performance (Zhang et al.,

2025b; Qi et al., 2024; Ma et al., 2025). In light of these advances and the observed instability of policies across stochastic conditions, we aim to achieve meaningful performance gains by allocating inference-time computation.

The inference-time sampling framework as shown in Figure 1 mainly contains two key components. (1) **Candidate Sampling:** strategy for generating multiple candidates, and (2) **Candidate Verification:** method for constructing a verifier that evaluates candidate actions. High-quality candidate sampling is necessary to ensure that plausible actions are available, while effective candidate verification is essential to select a reliable action among them. For candidate sampling, existing methods mainly perform action sampling from policy models (Nakamoto et al., 2024; Yang et al., 2025a), which increases the computational cost by an order of magnitude. While sampling from a Gaussian distribution (Kwok et al., 2025) is efficient, it neglects inter-dimensional correlations of action chunks and therefore yields candidates that deviate from the true action distribution. For candidate verification, the absence of off-the-shelf models capable of understanding and interpreting low-level actions poses a critical challenge.

In this paper, we propose a plug-and-play inference-time sampling framework for generalist robotic policies, named TapSampling. For action sampling, we employ an Action-VAE to model the intra-chunk correlations among action dimensions, which encodes initial actions into a compressed posterior distribution over latent, from which additional latent samples are drawn and decoded into action candidates. Benefiting from the learned low-dimensional posterior, the generated actions remain closer to the true action distribution. For action verification, we introduce a verifier that interprets low-level actions by predicting their impact on task progress. The verifier is trained with the intrinsic sequential characteristic of the robotic trajectory, without an additional data synthesis procedure. Furthermore, the predicted values with explicit semantics enable interpretable action selection.

The contributions are summarized as follows:

- **Policy-agnostic inference-time framework:** We propose a plug-and-play framework to improve the performance of pretrained generalist policies without further fine-tuning them.

- **Action sampling with learned posterior:** We introduce an Action-VAE that encodes actions into a compressed distribution, from which more action candidates in proper modes can be sampled efficiently.

- **Action selection via task progress verification:** We introduce a task-progress-understanding verifier that assigns values with explicit semantics for query actions, enabling interpretable action selection.

- Experiments demonstrate that TapSampling enhances the manipulation capabilities of diverse representative policies in both simulated and real-world settings.

## 2. Related Works

### 2.1. Generalist Robotic Policy

Developing generalist robotic policies that can execute a wide variety of tasks following different instructions is an important research direction of embodied control. The recent success of video foundation models (Blattmann et al., 2023; Yang et al., 2025b) and VLMs (Karamcheti et al., 2024; Beyer et al., 2024) has spurred a flurry of interest in building generalist robotic policies based on pretrained foundation models. They typically extend the vocabulary of VLMs and generate actions through next-token prediction (Kim et al., 2024), or take the generated tokens, images, videos, or immediate features of foundation models as the conditioning signal, and generate actions through action expert modules (Black et al., 2025b; Bu et al., 2024; Wen et al., 2025; Hu et al., 2025; Tian et al., 2025). As data and model sizes scale up, policy models demonstrate excellent performance. Despite these advances, current policies typically commit to a single action at each decision step (single-shot inference), resulting in no mechanism to correct failures caused by stochastic variance and potentially producing unstable behavior at deployment (Yang et al., 2025a). In this paper, we investigate inference-time sampling as a complementary axis to improve performance without additional policy fine-tuning.

### 2.2. Inference-Time Sampling

**Inference-Time Sampling for LLMs and Diffusion Models.** LLMs have achieved remarkable success in producing diverse, high-quality answers to the same question with the autoregressive next-token prediction paradigm. In addition to scaling training data and model size, inference-time sampling has emerged as a prominent research focus (Zhang et al., 2025b). Recent works have demonstrated that repeated or structured inference—e.g., chain-of-thought prompting (Wei et al., 2022; Lightman et al., 2024), multiple-sample self-consistency (Wang et al., 2023; Guangya Wan, 2025), and verifier-guided selection (Li et al., 2023; Zhao et al., 2025)—can improve output quality for high-level, semantic tasks. In parallel, diffusion models on the vision side are sensitive to initial noise and have benefited from techniques that optimize or search initial noises (Zhou et al., 2025; Tong et al., 2025; Guo et al., 2024), or that search samples or sampling paths with auxiliary verifiers (Ma et al., 2025). Crucially, these domains benefit from off-the-shelf evaluators (e.g. VLMs) and semantic metrics (e.g., Inception Score), which make verification tractable. In contrast, robotic policies emit low-level actions whose

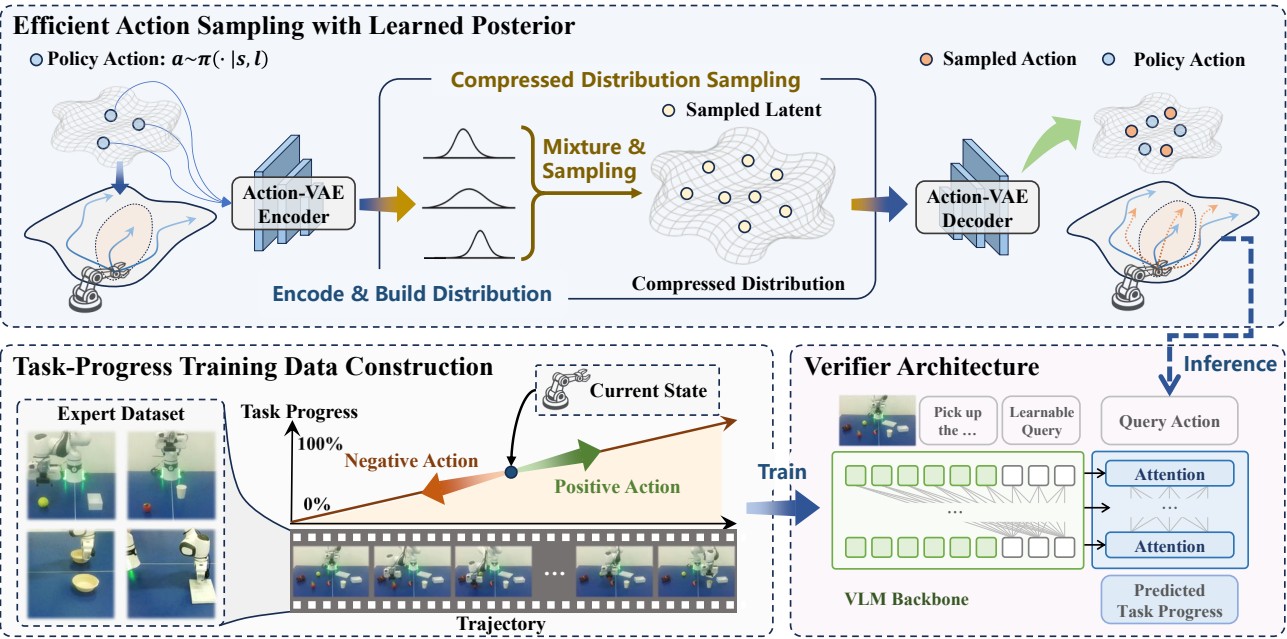

*Figure 2.* **TapSampling overview**. For action sampling, a small set of actions is sampled from the policy, encoded and mixed into a compressed latent distribution by the Action-VAE encoder. Multiple latent samples are then drawn from the learned posterior and decoded into diverse, high-quality action candidates efficiently. For action verification, positive and negative training examples are constructed automatically from expert trajectories using their intrinsic sequential information, and a verifier is trained to predict task-progress changes, which enables interpretable action selection.

quality must be judged by their physical outcomes, making off-the-shelf verification far less straightforward.

**Inference-Time Sampling for Robotic Policies.** Recently, inference-time sampling has begun to attract attention in robotics. Existing methods, however, rarely study principled action sampling strategies. A common practice is to repeatedly sample actions from the policy (Nakamoto et al., 2024; Yang et al., 2025a), which can substantially increase inference latency. RoboMonkey (Kwok et al., 2025) constructs a Gaussian distribution from a small set of actions produced by the policy to sample additional candidates. However, such approaches ignore intra-action correlations across dimensions and therefore tend to generate samples that deviate from the action distribution. On the verification side, recent work trains verifiers through hand-crafted function (Liu et al., 2025), offline reinforcement learning (Nakamoto et al., 2024), preference learning (Kwok et al., 2025; Dai et al., 2025), or state-count-based optimization (Yang et al., 2025a). They typically produce scores with limited interpretability or are tightly coupled to specific policy architectures, which limits their plug-and-play applicability across different policies.

### 2.3. Task Progress Understanding

Recent work has incorporated task-understanding models as value or reward functions in embodied reinforcement learn-

ing (RL). One line of research employs pretrained VLMs to determine whether a task has been completed and uses that signal as a reward (Du et al., 2023; Yang et al., 2024; Venkataraman et al., 2025; Wang et al., 2024). Another approach, more closely aligned with our method, explicitly predicts task progress. They estimate task progress changes (Zhai et al., 2025; Zhang et al., 2025a) or steps-to-completion (Ghasemipour et al., 2025; Intelligence, 2025) for PPO-based RL, REINFORCE, and offline RL, respectively. However, such value/reward functions cannot be employed for inference-time sampling because they evaluate progress solely from **current state** and do not model the effects of candidate low-level actions. In contrast, our verifier predicts task progress changes conditioned on the query actions, estimating their expected **future outcome** and thereby providing interpretable scores for action selection at inference time.

## 3. Inference-Time Sampling

### 3.1. Overview

As shown in Figure 2, our proposed TapSampling achieves inference-time sampling via two key components: (1) Efficient Action Sampling with Learned Posterior and (2) Interpretable Action Selection via Task Progress Verification. Specifically, we first train a low-dimensional posterior through action reconstruction. Building upon the learned

posterior, we construct a compressed latent distribution with a small set of policy actions, from which an arbitrary number of actions can be sampled (Sec. 3.2). Subsequently, we construct training data for task-progress prediction with the intrinsic sequential information of robotic trajectories, and train a task-progress-understanding verifier for action selection (Sec. 3.3).

## 3.2. Action Sampling with Learned Posterior

Action sampling plays a critical role in inference-time strategies. Existing methods typically either draw multiple samples directly from the policy model, which substantially increases inference latency, or sample from a Gaussian distribution, which ignores correlations among action dimensions. As an optional sampling strategy, we introduce an Action-VAE that models a learned, low-dimensional posterior, providing a flexible mechanism to trade off sampling efficiency and action quality.

**Approximate Posterior Learning.** The Action-VAE adopts a transformer-based architecture (Chen et al., 2023) for its encoder and decoder. During training, the encoder $\mathcal{E}$ takes as input an action chunk $a$ and compresses it into a latent Gaussian distribution with mean $\mu$ and diagonal covariance $\mathrm{diag}(\sigma^2)$, from which a low-dimensional latent $z$ is sampled, while the decoder $\mathcal{D}$ aims to reconstruct the original action from the latent:

$$q_{\mathcal{E}}(z \mid a) = \mathcal{N}\big(z; \mu_{\mathcal{E}}(a), \mathrm{diag}(\sigma_{\mathcal{E}}^2(a))\big),$$
$$z \sim q_{\mathcal{E}}(z \mid a), \tag{1}$$
$$\hat{a} = \mathcal{D}(z).$$

Specifically, $\mathcal{E}$ learns an approximate posterior distribution over the latent variables, conditioned on the input action. The Action-VAE is trained with a reconstruction loss and a Kullback–Leibler (KL) regularization term that penalizes the divergence between the learned posterior distribution and a standard normal prior:

$$\mathcal{L}_{avae} = \mathcal{L}_{rec} + \lambda_{\mathcal{KL}}\mathcal{L}_{\mathcal{KL}} \tag{2}$$

**Latent Space Sampling.** During inference, given a set of actions $A_{\pi} = \{a_i\}_{i=1}^{N}$ sampled from the policy model $\pi(a \mid s, l)$, where $s$ and $l$ denote the current state and task description, respectively, we aim to generate multiple action candidates that approximately follow the policy distribution with the Action-VAE. Specifically, we first encode each policy action into a compressed posterior distribution, obtaining a set of latent distributions:

$$\big\{\, q_{\mathcal{E}}(z \mid a_i) \,\big\}_{i=1}^{N} = \big\{\mathcal{N}\big(z; \mu_i, \mathrm{diag}(\sigma_i^2)\big)\big\}_{i=1}^{N}, \tag{3}$$

where $\mu_i = \mu_{\mathcal{E}}(a_i)$ and $\sigma_i^2 = \sigma_{\mathcal{E}}^2(a_i)$ are the outputs of the encoder $\mathcal{E}$. Following this, the mixed posterior distribution

is computed as:

$$q_{\mathrm{mix}}(z \mid A_{\pi}) = \frac{1}{N}\sum_{i=1}^{N} q_{\mathcal{E}}(z \mid a_i) \tag{4}$$

from which an arbitrary number of latent samples can be efficiently drawn and decoded into action candidates:

$$z_i \sim q_{\mathrm{mix}}(z \mid A_{\pi}),$$
$$A^* = \{\mathcal{D}(z_i)\}_{i=1}^{M}. \tag{5}$$

By learning a compressed posterior that captures the underlying patterns of real actions, the Action-VAE enables efficient generation of diverse action candidates from a small set of policy-sampled actions while maintaining close adherence to the original policy distribution.

It is notable that the Learned Posterior Sampling is not intended to achieve better performance compared with the Policy Sampling strategy. Instead, we propose it as a practical alternative that significantly reduces inference latency (about 5× faster than Policy Sampling) and achieves a favorable trade-off between efficiency and effectiveness. We validate these claims empirically in Section 4.3.

## 3.3. Action Selection via Task Progress Verification

A central challenge in inference-time sampling is constructing a value function capable of scoring candidate actions. Given the current state $s$ and task description $l$, the value function $\mathcal{V}$ is expected to produce a score $q$ for a candidate action $a$:

$$q = \mathcal{V}(s, l, a). \tag{6}$$

People are excellent discriminators due in part to their rich background knowledge and experience. These endow humans with strong task-understanding abilities, allowing them to generalize judgments of task progress even for novel tasks. Inspired by this, we train a verifier that evaluates the progress of task completion.

**Training Data Construction.** In imitation learning, trajectories in the dataset typically comprise an instruction along with a sequence of states and actions: $\tau = \{l, (s_1, a_1), (s_2, a_2), \dots, (s_t, a_t)\}$. Each trajectory exhibits an intrinsic sequential structure. Under the assumption that task progress grows linearly over time, we assign a task progress value $p_i = i/t$ to each timestep $i$, yielding a trajectory of the form: $\tau = \{l, (s_1, a_1, p_1), (s_2, a_2, p_2), \dots, (s_t, a_t, p_t)\}$, where the task progress $p$ increases gradually from $1/t$ to 1.

Given a random state $s_i$ from the dataset, a positive training sample can be constructed as:

$$(l, s_i, a_{i:i+k-1}) \mapsto (p_{i+k} - p_i) = \frac{k}{t}, \tag{7}$$

*Table 1.* **Main results on the CALVIN ABC→D benchmark.** TapSampling significantly improves the task success rate and the average success length of representative non-deterministic policies (Diffusion Policy, OpenVLA, and VPP) in a plug-and-play manner without further fine-tuning these policies.

| Method | $i^{th}$ Task Success Rate | | | | | Avg. Len ↑ |
|---|---|---|---|---|---|---|
| | **1** | **2** | **3** | **4** | **5** | |
| Robo-Flamingo (Li et al., 2024) | 82.4 | 61.9 | 46.6 | 33.1 | 23.5 | 2.48 |
| RoboDual (Bu et al., 2024) | 94.4 | 82.7 | 72.1 | 62.4 | 54.4 | 3.66 |
| ReconVLA (Song et al., 2026) | 95.6 | 87.6 | 76.9 | 69.3 | 64.1 | 3.95 |
| Seer (Tian et al., 2025) | 96.3 | 91.6 | 86.1 | 80.3 | 74.0 | 4.28 |
| DreamVLA (Zhang et al., 2025c) | **98.2** | **94.6** | **89.5** | 83.4 | 78.1 | 4.44 |
| Diffusion Policy (Chi et al., 2023) | 82.1 | 61.7 | 45.6 | 31.4 | 20.5 | 2.41 |
| + TapSampling | 83.9 (+1.8) | 65.1 (+3.4) | 48.8 (+3.2) | 35.7 (+4.3) | 24.7 (+4.2) | 2.58 (+0.17) |
| OpenVLA (Kim et al., 2024) | 93.4 | 78.2 | 64.1 | 52.2 | 42.4 | 3.30 |
| + TapSampling | 94.5 (+1.1) | 80.9 (+2.7) | 68.6 (+4.5) | 57.7 (+5.5) | 48.8 (+6.4) | 3.51 (+0.21) |
| VPP (Hu et al., 2025) | 96.4 | 92.3 | 88.4 | 84.0 | 78.3 | 4.39 |
| + TapSampling | 96.5 (+0.1) | 92.9 (+0.6) | 89.4 (+1.0) | **86.4** (+2.4) | **81.1** (+2.8) | **4.46** (+0.07) |

indicating that the action sequence $a_{i:i+k-1}$ produces a positive increment in task progress $\frac{k}{t} > 0$ when executed from the current state $s_i$. Conversely, a corresponding negative sample can be defined as:

$$(l, s_i, a^r_{i:i-k+1}) \mapsto (p_i - p_{i-k}) = \frac{-k}{t}, \qquad (8)$$

representing that the reversed action sequence $a^r_{i:i-k+1}$ results in a negative change in task progress $\frac{-k}{t} < 0$, and thus impedes task completion.

Specifically, actions are typically represented by the robot's joint angles or end-effector pose, together with the gripper state. Given this semantically meaningful action representation, reversed actions can be directly derived from the dataset. Crucially, our method constructs positive and negative samples from expert trajectories to train a model that predicts task-progress changes, without requiring any additional manual annotation. This design allows the approach to be seamlessly applied to most existing robotic datasets.

**Verifier Architecture.** The capability of understanding visual observations and text instructions is essential for the verifier. To this end, we build the verifier upon VLA-Adapter (Wang et al., 2026), a state-of-the-art VLA architecture that employs Qwen2.5-0.5B (Team et al., 2024) as its backbone for processing visual input and instructions, incorporates learnable queries for feature extraction, and generates actions through an action head. As shown in Figure 2, we leverage the same backbone to extract features, modify the action head to take the query actions as input, and predict corresponding task-progress changes. We formulate the task-progress prediction as a regression problem and optimize the model with L1 loss:

$$\mathcal{L}_{tap} = \|\mathcal{V}(s, l, a) - \Delta p\|_1 \qquad (9)$$

**Action Selection in Inference-time.** A key advantage of

our approach is that the predicted scores carry a concrete semantic meaning, reflecting their expected impact on task progress. This property enables interpretable action selection. Specifically, a negative score (e.g., $-10\%$) indicates that the action would impede task execution. Small positive values (e.g., $10\%$) generally correspond to stable, reliable execution, whereas larger positive values (e.g., $20\%$) indicate more rapid task execution. During inference, actions with scores below a predefined threshold are first discarded, and a score-weighted average of the remaining candidates is computed to produce a single high-quality, stable action. The algorithmic description is shown in the Appendix B.

**Inference Efficiency.** At inference time, all candidates are generated from the same observation–instruction pair, which allows our verifier to evaluate them in parallel. Specifically, the VLM backbone is executed only once to obtain hidden states that are shared across all candidates. These hidden states are then replicated into a batch and passed through the lightweight action head to predict task-progress scores for all query actions simultaneously.

## 4. Experiments

In this section, to evaluate TapSampling and validate the contribution of each component, experiments on simulated (CALVIN, LIBERO) and real-world environments are designed to answer three key questions:

***Q1***. Can TapSampling improve the performance of existing policies through inference-time sampling across diverse environments and tasks?

***Q2***. How does the sampling strategy affect the inference latency and the quality of action candidates?

***Q3***. How does the task-progress-understanding verifier af-

*Table 2.* **Results on the LIBERO-Long benchmark.** Our method improves the success rate of representative VLA, $\pi_{0.5}$, from 96.8% to 98.0%. Results marked with † indicate that we directly follow the official results in TACO (Yang et al., 2025a).

| Task ID | $\pi_{0.5}$† | + TACO† | $\pi_{0.5}$ | + Ours |
|---|---|---|---|---|
| 0 | 98.0 | **100.0** | 98.0 | 98.0 (0.0) |
| 1 | **100.0** | 96.0 | 98.0 | 96.0 (-2.0) |
| 2 | 98.0 | 98.0 | 100.0 | 100.0 (0.0) |
| 3 | 98.0 | 100.0 | 100.0 | 100.0 (0.0) |
| 4 | 98.0 | 98.0 | 96.0 | **100.0** (+4.0) |
| 5 | 100.0 | 100.0 | 96.0 | 100.0 (+4.0) |
| 6 | 96.0 | 92.0 | 94.0 | **98.0** (+4.0) |
| 7 | 94.0 | 100.0 | 100.0 | 100.0 (0.0) |
| 8 | 68.0 | 86.0 | **92.0** | 90.0 (-2.0) |
| 9 | 98.0 | 96.0 | 94.0 | 98.0 (+4.0) |
| Avg | 94.8 | 96.6 | 96.8 | **98.0** (+1.2) |

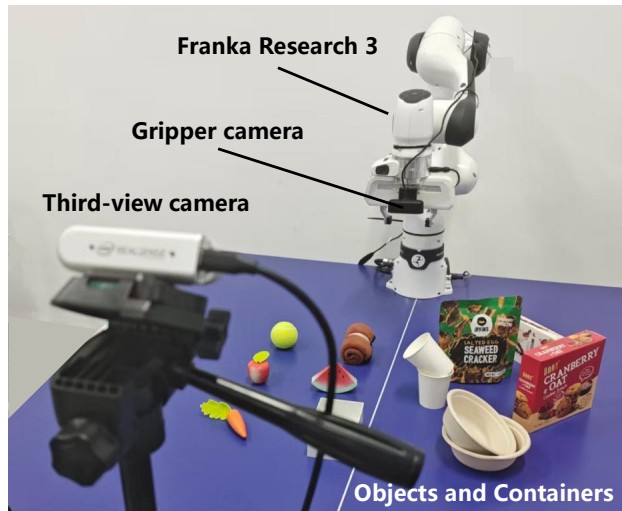

*Figure 3.* Real-world experimental environment.

fect the number of steps required to complete a task?

### 4.1. Simulation Setups and Baselines

**Simulation benchmark.** We evaluate the TapSampling framework on the CALVIN and LIBERO benchmarks.

- **CALVIN** (Mees et al., 2022): A benchmark designed for learning long-horizon, language-conditioned robot manipulation policies. In the ABC→ D setting, both policies and our verifier are trained in environment A, B, and C, and evaluated in unseen environment D. Policies with a higher average success length are considered to exhibit superior zero-shot, long-horizon manipulation capabilities.

- **LIBERO** (Liu et al., 2023a): A benchmark for life-long learning in decision-making. Following (Yang et al., 2025a), we focus on the most challenging suite, LIBERO-Long, as previous works have already achieved near-perfect success rates on the other suites.

**Baseline.** We select a subset of representative generalist robot policies and combine them with TapSampling.

- **Diffusion Policy** (Chi et al., 2023): We implemented the diffusion policy with advanced components in the CALVIN benchmark: DinoV2 visual encoder, Q-Former encoder, and DiT action expert.

- **OpenVLA** (Kim et al., 2024): A representative VLA model that generates discrete action tokens autoregressively in a next-token-prediction fashion. We adopt the action-chunk variant of OpenVLA from (Bu et al., 2024) when evaluating on the CALVIN benchmark.

- **VPP** (Hu et al., 2025): A representative policy that extracts dynamic features from a pretrained video diffusion model and generates action chunks with DiT action expert. We utilize the official checkpoints in the CALVIN benchmark.

- **$\pi_{0.5}$** (Black et al., 2025a): A representative VLA model with a flow matching action expert. We employ community-developed open-source weights in the LIBERO benchmark.

**Quantitative Results.** Evaluation on the CALVIN benchmark is shown in Table 1. The results of Robo-Flamingo, RoboDual, ReconVLA, Seer, and DreamVLA are taken from their official reports and are included to illustrate the performance of state-of-the-art methods on the CALVIN benchmark. TapSampling yields consistent performance improvements for Diffusion Policy (from 2.41 to 2.58), Open-VLA (from 3.30 to 3.51), and VPP (from 4.39 to 4.46). Furthermore, results in Table 2 indicate that our method demonstrates performance gains for $\pi_{0.5}$ from 96.8% to 98.0% on LIBERO-Long. Overall, the results show that TapSampling is effective at improving performance across diverse environments, tasks, and policies.

### 4.2. Real-world Experiments

**Environment Setting.** Experiments are conducted using a 7-DoF Franka Research 3 robotic arm equipped with a 1-DoF gripper. Models receive images from the third-view camera and the wrist camera as input. We evaluate our method across three categories of manipulation tasks: (1) Knock down <object>, which primarily assesses the visual perception ability; (2) Put <object> into <container>, which focuses on object grasping and placement accuracy; and (3) Stack <object>, which requires precise spatial alignment

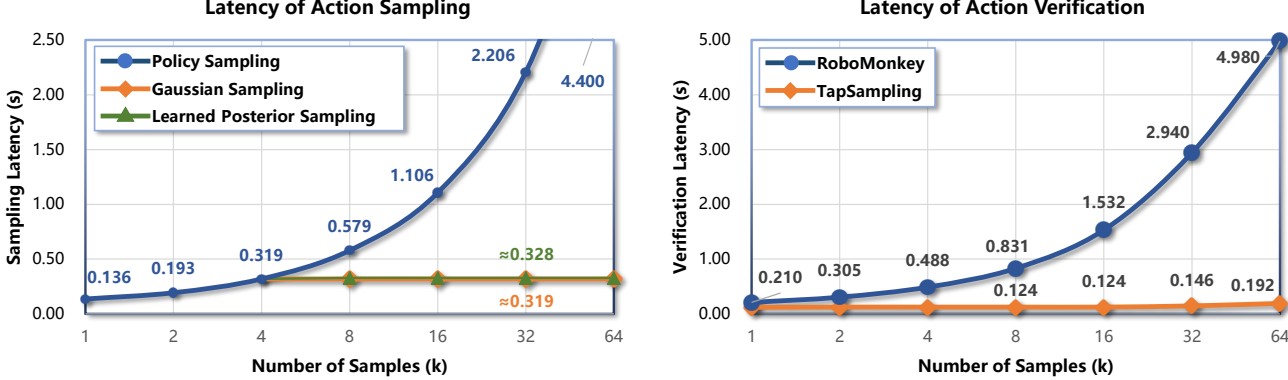

*Figure 4.* (**Left**) **Latency of Action Sampling.** We report the average latency across sampling strategies. Policy Sampling significantly increases the latency while Gaussian Sampling and our strategy incur negligible additional latency (less than 0.01 s) given a small set ($n = 4$) of initial actions generated by the policy model. (**Right**) **Latency of Action Verification.** Thanks to its architecture, TapSampling efficiently evaluates multiple candidates, achieving an approximately 12× speedup over RoboMonkey when $k = 16$.

*Table 3.* **Results on the real-world tasks.**

| Method | Knock Down | | Pick and Place | | Stack | | Average |
|---|---|---|---|---|---|---|---|
| | Seen | Unseen | Seen | Unseen | Seen | Unseen | |
| $\pi_0$ | 90.0 | 86.7 | 73.3 | 65.0 | 80.0 | 75.0 | 78.3 |
| +TapSampling | **93.3** (+3.3) | **93.3** (+6.6) | **76.7** (+3.4) | **66.7** (+1.7) | **85.0** (+5.0) | **85.0** (+10.0) | **83.3** (+5.0) |

and contact-aware placement. We collect 100 successful demonstration trajectories for each task to fine-tune the policy model $\pi_0$ and TapSampling.

**Quantitative Results.** The success rates in real-world experiments are shown in Table 3. TapSampling improves the average success rates of $\pi_0$ from 78.3% to 83.3%. More importantly, our approach allows for greater adjustment of the end-effector position before grasping and placing, resulting in more precise manipulation.

### 4.3. Further Analysis

**Sampling Efficiency Analysis.** We answer the ***Q2: how does the sampling strategy affect the inference latency and the quality of action candidates*** to evaluate our proposed sampling strategy. While the inference-time sampling framework improves policy performance by generating multiple candidate actions, it inevitably introduces additional computation. We therefore propose an optional sampling strategy, Learned Posterior Sampling, which efficiently produces an arbitrary number of candidates from a small set of initial actions proposed by the policy. Figure 4 (Left) reports the average latency to generate $k$ candidate actions with three sampling strategies. Results demonstrate that directly sampling $k = 16$ candidate actions from the policy increases sampling latency by approximately 8×, substantially degrading inference efficiency. For Gaussian Sampling and our method, we utilize four actions from the policy to construct the distribution. Once this small set of actions is obtained, both strategies incur negligible additional latency

(less than 0.01 s) when generating more action candidates.

**Sampling Fidelity Analysis.** First, we evaluate how the sampling strategy affects performance. As shown in Table 4, Policy Sampling achieves the highest performance, consistent with the findings of previous work (Kwok et al., 2025), demonstrating the strong potential of our method. However, Policy Sampling significantly increases the inference latency by approximately 20×. In contrast, although efficient, Gaussian Sampling results in limited improvement, possibly because it generates candidates that deviate from the action distribution. Learned Posterior Sampling strikes a balance between efficiency and effectiveness. It achieves better performance, especially in long-horizon execution (e.g., tasks 4 and 5), compared with Gaussian Sampling. This could be attributed to Action-VAE's capability to generate actions that better align with the policy distribution.

For each of 1,000 random evaluation states, we sample 256 actions using each of Policy Sampling, Gaussian Sampling, and Learned Posterior Sampling to quantitatively assess distributional similarity. The Gaussian distribution and our compressed latent distribution are built upon the first 4 policy actions. The Maximum Mean Discrepancy (MMD) values, computed with multiple kernel bandwidths, between the actions generated by each method (Gaussian or Posterior) and the reference policy samples are reported in Table 5. Our strategy consistently yields lower MMD values, indicating a closer alignment with the policy distribution.

**Verification Efficiency Analysis.** We further compare the verification latency of TapSampling with another inference-

*Table 4.* **Ablation study on sampling strategies.** Learned Posterior Sampling yields a good balance between efficiency and effectiveness.

| Method | $i^{th}$ Task Success Rate | | | | | | Latency ↓ |
|---|---|---|---|---|---|---|---|
| | 1 | 2 | 3 | 4 | 5 | Avg. Len ↑ | |
| VPP (Hu et al., 2025) | 96.4 | 92.3 | 88.4 | 84.0 | 78.3 | 4.39 | 0.136 |
| Gaussian Sampling | 96.7 (+0.3) | 92.6 (+0.3) | 89.2 (+0.8) | 84.8 (+0.8) | 79.9 (+1.6) | 4.43 (+0.04) | 0.465 |
| Learned Posterior Sampling | 96.5 (+0.1) | 92.9 (+0.6) | 89.4 (+1.0) | 86.4 (+2.4) | 81.1 (+2.8) | 4.46 (+0.07) | 0.488 |
| Policy Sampling | **97.1** (+0.7) | **93.2** (+0.9) | **90.3** (+1.9) | **87.1** (+3.1) | **82.4** (+4.1) | **4.50** (+0.11) | 2.638 |

*Table 5.* Maximum Mean Discrepancy (MMD) comparison of different action sampling strategies across multiple bandwidths.

| Strategy | MMD (RBF kernel, $\gamma$) | | | | |
|---|---|---|---|---|---|
| | 2.0 | 4.0 | 6.0 | 8.0 | 10.0 |
| Gaussian | 0.098 | 0.155 | 0.187 | 0.204 | 0.212 |
| **Ours** | **0.064** | **0.082** | **0.095** | **0.104** | **0.112** |

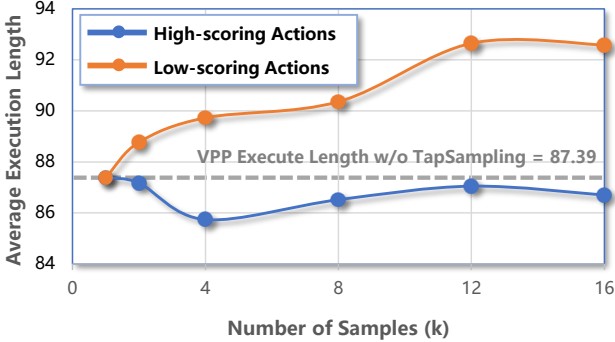

*Figure 5.* Comparison of average execution length when selecting high- and low-scoring actions.

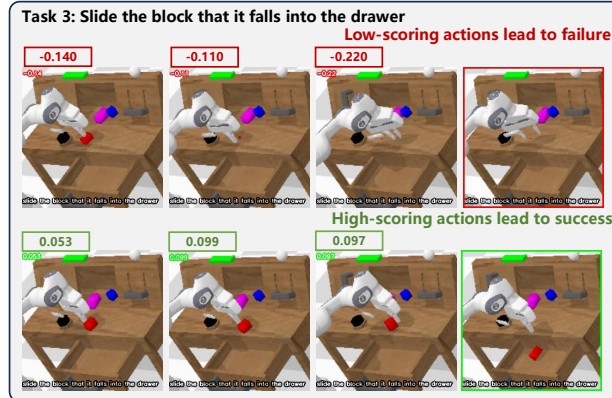

*Figure 6.* **Action verification examples.** Low-scoring actions lead to incorrect collisions that result in task failure, whereas high-scoring actions result in successful execution.

time sampling method in Figure 4 (Right). RoboMonkey (Kwok et al., 2025) utilizes LLaVA-7B (Liu et al., 2023b) as its backbone, replaces the final layer with a reward head, and trains the model as a verifier via preference learning. We measure the average verification latency for evaluating varying numbers of actions using their official implementation. Statistical results indicate that the verification latency of RoboMonkey increases significantly as the number of actions grows. In contrast, the VLM backbone in TapSampling (Qwen2.5-0.5B (Team et al., 2024)) is invoked only once per verification, and multiple candidate actions are evaluated in a batch through the lightweight action head to query task progress scores. Consequently, our method consistently maintains high efficiency.

**Verification Effectiveness Analysis.** We answer the *Q3: how does the task-progress-understanding verifier affect the number of steps required for task completion* to evaluate our proposed verifier. The model is trained to predict the impact of candidate actions on task progress. Ideally, selecting high-scoring actions allows the task to be completed in fewer steps, whereas choosing low-scoring actions can hinder or delay task execution. Specifically, at each

decision step, we sample $k$ actions from the policy model and execute either the highest-scoring or the lowest-scoring action. As shown in Figure 5, selecting low-scoring actions substantially increases the number of steps required for task completion, whereas choosing high-scoring actions leads to task completion in fewer steps.

We further provide a visual comparison in Figure 6. Selecting low-scoring actions results in an incorrect collision between the robotic arm and the object, which disrupts the scene and leads to task failure. In contrast, selecting high-scoring actions successfully completes the task. More examples are shown in the Appendix E.

## 5. Conclusion

In this paper, we investigate inference-time sampling as a complementary axis for improving generalist robotic policies, motivated by the instability of the single-shot inference paradigm. We propose **TapSampling**, a plug-and-play inference-time framework that improves policy performance without further fine-tuning the policies. TapSampling enables sampling-and-verification through a task-progress-understanding verifier and balances efficiency and performance by sampling from the learned posterior. Our findings suggest that allocating computation to inference time offers an effective and model-agnostic pathway for improving the reliability and performance of generalist robotic policies.

## Acknowledgements

This work was supported by the National Natural Science Foundation of China (Grant Nos. U2574212 and 62272134), the Natural Science Foundation of Shandong Province (Grant No. ZR2024LGY002), the Key R&D Program of Shandong Province (No. 2025CXGC010115), the Shenzhen Fundamental Research Program (JCYJ20250604145514018), the Guangdong Basic and Applied Basic Research Foundation (General Program, No. 2026A1515011557) and the NSFC Young Scientists Fund (No. 62506096).

## Impact Statement

This paper presents work whose goal is to advance the field of Machine Learning. There are many potential societal consequences of our work, none which we feel must be specifically highlighted here.

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

# A. Motivation: Instability of Non-deterministic Policy

Although existing non-deterministic generalist policies achieve remarkable performance, they exhibit noticeable instability, succeeding in some trials and failing in others under identical environmental conditions. We conduct experiments on the CALVIN benchmark (Mees et al., 2022) with VPP (Hu et al., 2025) to illustrate the instability. Specifically, the policy is permitted $n$ retry attempts upon failure during the execution of long-horizon task sequences. As shown in Table 6, VPP achieves an average success length of 4.39 without any retries. Allowing a single retry increases the average success length substantially to 4.61, with 39.5% of previously failed subtasks converted to successes upon retry. As $n$ rises to 4, the average success length converges to approximately 4.72, indicating that the model has a remarkable upper bound. These observations motivate us to explore inference-time strategies for improved action selection and for mitigating the uncertainty induced by stochastic generation, thereby improving the performance of policy models.

**Remaining Gap to the Retry Upper Bound.** Results from Retry-Attempt reveal substantial room for further improvement. This gap should be interpreted in light of the fundamentally different evaluation protocols. The Retry Upper Bound is more appropriately viewed as a trajectory-level pass@k result because it allows multiple independent full-trajectory attempts, relies on oracle success or failure signals from the simulator, and resets the environment to a clean initial state after each failure. In contrast, TapSampling is designed to improve pass@1 within a single continuous rollout. This distinction is particularly important for real-world deployment, where oracle resets are typically unavailable. This highlights the core motivation of our work, which is to trade inference-time computation for a higher pass@1 success rate.

*Table 6.* **Instability of non-deterministic policies.** Providing retry attempts to VPP significantly increases the average success length.

| Retry Attempt | $i^{th}$ Task Success Rate | | | | | |
|---|---|---|---|---|---|---|
| | 1 | 2 | 3 | 4 | 5 | Avg. Len ↑ |
| VPP (Hu et al., 2025) | 96.4 | 92.3 | 88.4 | 84.0 | 78.3 | 4.39 |
| $n = 1$ | 97.1 | 94.2 | 92.0 | 90.1 | 87.2 | 4.61 |
| $n = 2$ | 97.5 | 94.8 | 93.0 | 91.5 | 89.0 | 4.66 |
| $n = 3$ | **98.1** | **95.9** | 94.2 | 92.7 | 90.8 | 4.72 |
| $n = 4$ | 98.0 | 95.5 | 94.2 | **93.1** | **91.2** | **4.72** |

# B. Algorithmic Description

Algorithm 1 presents an algorithmic description of the sampling-verification-selection procedure of TapSampling.

---
**Algorithm 1:** Inference-time sampling framework, TapSampling

---
**Input:** Policy model $\pi(a \mid s, l)$, Action-VAE encoder $\mathcal{E}(\mu, \sigma^2 \mid a)$, Action-VAE decoder $\mathcal{D}(z)$, TapSampling verifier
$\quad\quad \mathcal{V}(s, l, a)$, initial state $s$, task instruction $l$, selection threshold $\theta$.

**Output:** Action chunk $a^\star$

Sample actions $A_\pi = \{a_i\}_{i=1}^N$ from the policy model: $a_i \sim \pi(a \mid s, l)$

Transform each action into distribution: $q_\mathcal{E}(z \mid a_i) = \mathcal{N}\big(z; \mu_\mathcal{E}(a_i), \mathrm{diag}(\sigma_\mathcal{E}^2(a_i))\big)$

Build a mixed posterior distribution: $q_{\mathrm{mix}}(z \mid A_\pi) = \frac{1}{N} \sum_{i=1}^N q_\mathcal{E}(z \mid a_i)$

Sample latents from the distribution: $z_i \sim q_{\mathrm{mix}}(z \mid A_\pi)$

Map the latents into actions: $A^\star = \{\mathcal{D}(z_i)\}_{i=1}^M$

Evaluation all actions: $q_i = \mathcal{V}(s, l, a_i)$

Retain actions with scores above the given threshold.: $\hat{A} = \{a_i \in \{A_\pi, A^\star\} \mid s_i > \theta\}$

**if** no action left $(\hat{A} = \varnothing)$ **then**
$\quad\mid$ Simply select the action with the highest score: $a^\star = \mathrm{argmax}_{a_i \in \{A_\pi, A^\star\}} \mathcal{V}(s, l, a_i)$
**else**
$\quad\mid$ Compute the weighted average of the retained actions as $a^\star$
**Return** $a^\star$

---

## C. Additional Environment Details

Figure 7 illustrates the simulation benchmarks and the real-world tasks. The CALVIN ABC→D setting features zero-shot, long-horizon manipulation across 34 different tasks. Policies are evaluated by executing 1,000 preset task sequences. In each sequence, the policies are required to complete five subtasks sequentially. LIBERO-Long consists of 10 diverse tasks for evaluation. Following (Yang et al., 2025a), each task is evaluated over 50 trials.

In the real-world experiments, third-view and gripper images are collected at 25 FPS and interpolated to 50 FPS, while joint angles, gripper states, and end-effector poses are recorded at 50 Hz. Following (Black et al., 2025a), the policy controls the joint angles and gripper, resulting in an 8-DoF state and action space.

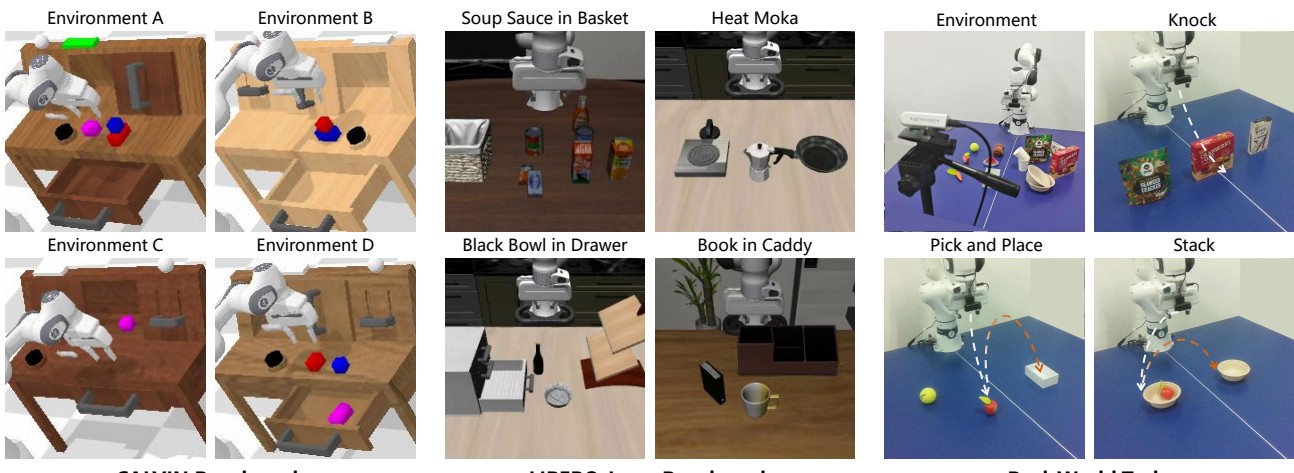

*Figure 7.* Illustration of the simulation benchmarks and real-world tasks.

## D. Training Details

We employ the action-chunk variant from (Bu et al., 2024) as OpenVLA (CALVIN), the official checkpoint from (Hu et al., 2025) for VPP (CALVIN), and community-developed open-source weights for $\pi_{0.5}$ (LIBERO) [1]. The training process of the Diffusion Policy (CALVIN), $\pi_0$ (Real-World), and the TapSampling verifier follows the settings listed in Table 7.

*Table 7.* Training configurations of policy models and TapSampling verifier models.

| Name | DP (CALVIN) | $\pi_0$ (Real-World) | TapSampling Verifier | | |
|---|---|---|---|---|---|
| | | | CALVIN | LIBERO | Real-World |
| Training Step | 50000 | 50000 | 65000 | 60000 | 20000 |
| LoRA Fine-tune | - | - | ✓ | ✓ | ✓ |
| LoRA Rank | - | - | 64 | 64 | 64 |
| LoRA Alpha | - | - | 128 | 128 | 128 |
| Optimizer | AdamW | AdamW | AdamW | AdamW | AdamW |
| Learning Rate Scheduler | CosineDecay | CosineDecay | CosineDecay | CosineDecay | CosineDecay |
| Maximun Learning Rate | $1 \times 10^{-4}$ | $1 \times 10^{-4}$ | $1 \times 10^{-5}$ | $1 \times 10^{-5}$ | $1 \times 10^{-5}$ |
| Batch Size | 48 | 32 | 16 | 16 | 16 |
| Input Action Shape | - | - | $10 \times 7$ | $10 \times 7$ | $50 \times 8$ |
| Output Shape | $10 \times 7$ | $50 \times 8$ | 1 | 1 | 1 |

---

[1] https://github.com/TensorAuto/OpenTau

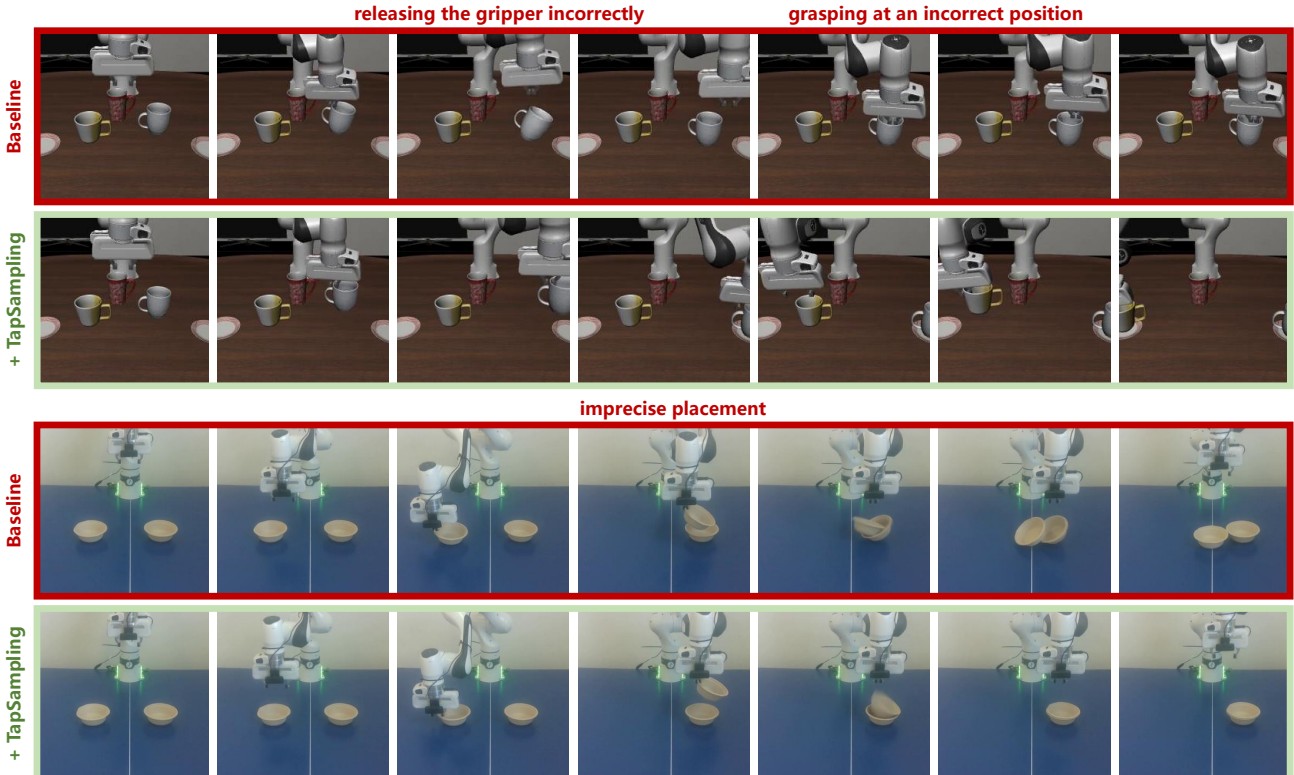

*Figure 8.* Additional examples in the LIBERO-Long benchmark and the real-world environment.

## E. Additional Examples

We provide more results in the LIBERO-Long and the real-world environment in Figure 8. Baseline policies occasionally produced incorrect gripper releases, misaligned grasps, or imprecise placements. When integrated with the TapSampling framework, these models successfully complete the tasks.

## F. Performance on OOD Actions

Reversed actions do not cover all semantically invalid actions. However, our goal is not to exhaustively model all failure modes, but to provide a contrastive signal that separates expert-like actions from poor candidates. We evaluated the verifier (trained on CALVIN ABC) on environment D with official demonstrations by scoring five types of actions to assess its robustness on out-of-distribution actions: (1) forward actions from expert data, (2) reversed actions (backward), (3) half-speed forward actions, (4) random action chunks from expert data, and (5) gripper-corrupted actions. As shown in Table 8, forward and reversed actions receive the highest and lowest average scores, respectively. Half-speed actions score roughly half as much as forward actions. Random and corrupted have average scores close to zero. Approximately 80% of bad actions are filtered with a threshold of 0.08. The verifier shows a certain degree of generalization to OOD actions.

## G. Ablation on Number of Samples

Ablation results with respect to the number of samples are reported in Figure 9. Specifically, we sample four candidate actions from the policy model and draw an additional $n - 4$ actions from the learned posterior distribution. The results demonstrate that TapSampling consistently improves the average success length of the baseline policy.

## H. Latent Dimension Analysis

Within an action chunk, actions at consecutive timesteps exhibit temporal coherence, resulting in strong intrinsic correlations. We encode action chunks into a compressed latent space using the Action-VAE, which captures underlying action patterns and improves the alignment of sampled actions with the true action distribution. Figure 9 shows that compressing action

*Table 8.* **TapSampling-Predicted Task Progress for Different Action Types.**

| Progress | Percentage (%) | | | | |
|---|---|---|---|---|---|
| | Forward | Backward | Half-speed | Random | Corrupted |
| $[-0.20, -0.16)$ | 0.2 | 8.1 | 0.1 | 0.9 | 7.7 |
| $[-0.16, -0.12)$ | 1.3 | 26.4 | 1.2 | 7.4 | 11.9 |
| $[-0.12, -0.08)$ | 2.4 | 22.9 | 2.9 | 16.0 | 11.8 |
| $[-0.08, -0.04)$ | 4.5 | 12.5 | 6.2 | 15.8 | 10.7 |
| $[-0.04,\ \ 0.00)$ | 8.2 | 9.3 | 8.7 | 12.4 | 10.1 |
| $[\ \ 0.00,\ \ 0.04)$ | 11.0 | 7.1 | 14.7 | 12.8 | 12.1 |
| $[\ \ 0.04,\ \ 0.08)$ | 13.1 | 5.7 | 34.2 | 12.2 | 11.6 |
| $[\ \ 0.08,\ \ 0.12)$ | 26.1 | 4.1 | 27.0 | 11.0 | 11.2 |
| $[\ \ 0.12,\ \ 0.16)$ | 24.7 | 2.5 | 4.5 | 9.9 | 7.2 |
| $[\ \ 0.16,\ \ 0.20)$ | 7.9 | 0.7 | 0.5 | 1.7 | 2.8 |
| Mean Progress | 0.0771 | -0.0713 | 0.0477 | -0.0020 | -0.0198 |

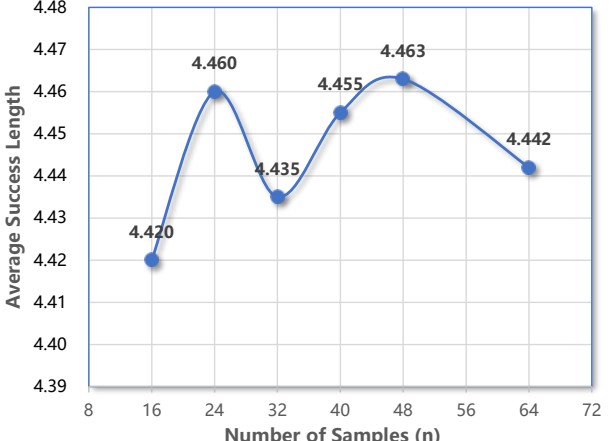 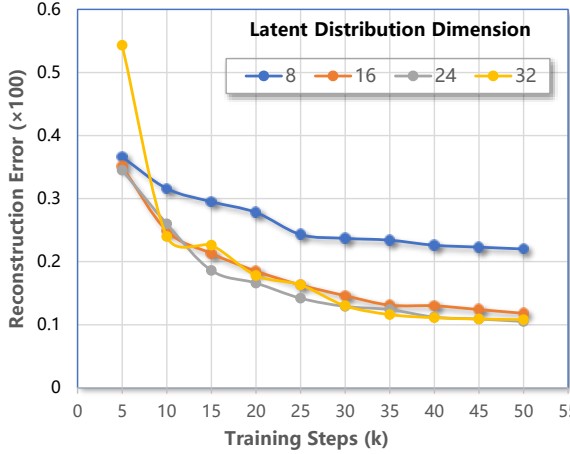

*Figure 9.* (Left) Ablation study on the number of samples. (Right) Effect of latent space dimensionality on reconstruction error.

chunks of shape $(10 \times 7)$ into 16, 24, or 32 dimensions maintains comparably low reconstruction loss on the validation set. Reducing the dimensionality to 8, however, incurs a marked increase in loss. Empirically, we adopt a 24-dimensional latent space for sampling and decoding.

## I. Discussion and Limitations

**Linear Task Progress Assumption.** True task progress in some tasks (e.g., tasks with long preparation phases) is not exactly linear. However, for inference-time sampling, the key requirement is not a perfectly calibrated global progress predictor, but a useful action-scoring signal that helps separate poor candidates from promising ones in the same environment state. In our method, the normalized-time target is intended as a practical continuous self-supervision signal for the action-conditioned verifier, rather than as a claim that physical task progress is strictly linear. Additionally, our supervision is constructed from expert demonstrations, and the labels still reflect which kinds of actions successful expert trajectories tend to take to complete the task, even when true progress is not exactly linear.

**Limitations.** TapSampling is designed to enhance pretrained policies via better action selection and thus remains bounded by the performance of the underlying policy. If the policy model lacks the capacity to generate correct actions, the performance of TapSampling is also limited. Furthermore, our experimental results show that TapSampling yields larger improvements on long-horizon tasks, such as Tasks 4 and 5 in CALVIN from 84.0%/78.3% to 86.4%/81.1%, while providing limited gains when the base policy is already close to saturation, such as Task 1 of CALVIN from 96.4% to 96.5%.

