# OpenReview forum: "TapSampling: Inference-Time Sampling with a Task-Progress-Understanding Verifier for Robotic Manipulation"
_ICML.cc/2026/Conference — ICML 2026 regular_

### Official Review · Reviewer_gsdS · 2026-03-11

**Soundness:** 3
**Presentation:** 3
**Significance:** 3
**Originality:** 3
**Overall Recommendation:** 5
**Confidence:** 4

**Summary:**

The authors improve task success rates by employing a test-time sampling-and-verification framework. Specifically, they introduce an action-VAE to enable efficient sampling of a large number of candidates, and learn a progress-aware verifier to select actions with positive contributions for execution.

**Compliance With Llm Reviewing Policy:**

Affirmed.

**Final Justification:**

the response is acceptable.

**Key Questions For Authors:**

1) The VAE reconstruction error is 0.1, and a weighted average is performed in the algorithm pseudocode in the appendix. Will this reconstruction error affect the overall accuracy? I raise this point because the overall reconstruction loss of many VLA models after convergence is an order of magnitude smaller.
2) Regarding the generalization issue of the verifier: without large-scale pre-training, the verifier acts similarly to a Q-function in reinforcement learning and may be fooled by non-expert OOD actions with poor performance. Could the authors elaborate on the potential generalization issues?
3) Could the authors provide ablation studies regarding the selection of the threshold parameter \theta ?
4) Any opensource plan if accepted?

If all the raised questions can be addressed, I am willing to increase the paper's score.

**Limitations:**

see questions

**Strengths And Weaknesses:**

Strengths：
1) The proposed method is sound and well-motivated.
2) The presentation is clear and well-organized
3) The work targets the critical issue that inherent stochasticity in sampling-based policy decisions degrades final performance, serving as a complementary improvement for modern decision models such as VLAs
4) Although test-time optimization via sampling has been studied before, the authors introduce an action-VAE and a verifier to reduce inference latency, which provides a practical contribution to the community.
5) The paper evaluates on standard benchmarks with sufficient experiments and detailed supplementary materials.

Weaknesses:
1) The overall idea is not highly novel. Learning a verifier for action selection has been widely explored. The core novelty mainly lies in using VAE for posterior modeling to accelerate sampling and reduce latency.
2) The verifier may suffer from OOD issues, as it is trained on task-progress data from expert demonstrations (positive vs. negative actions), which may not generalize well to newly sampled actions.
3) The method lacks theoretical analysis and guarantees.

---

> ### Author Rebuttal · Authors · 2026-03-31
>
> We thank the reviewer for the insightful comments and valuable feedback. We respond to the concerns below.
>
> > **[w1]** The overall idea is not highly novel. Learning a verifier for action selection has been widely explored. The core novelty mainly lies in using VAE for posterior modeling to accelerate sampling and reduce latency.
>
> Our contribution is mainly at the **framework level**: we formulate an inference-time sampling framework for frozen robot policies, in whichthe Action-VAE and the Action-Scorer are assigned roles that are distinct from their conventional uses in prior work or trained with a different strategy.
>
> **What is new about the verifier?** Our verifier is not a generic reward model or Q-function for policy optimization. It is an action-conditioned task-progress scorer used directly for inference-time evaluation of candidate actions. Its supervision is derived from the temporal structure of successful demonstrations, without reward labels, preference annotations, or offline RL. This gives the score explicit progress-related semantics.
>
> **Where does the novelty of TapSampling lie?** The main novelty of TapSampling is not the isolated use of a verifier or a VAE, but their combination into a plug-and-play inference-time framework that improves a frozen policy through local candidate expansion and action-conditioned scoring, without policy fine-tuning. We will revise the paper to make these distinctions from prior verifier-based action selection methods more explicit.
>
> > **[w2q2]** Generalization on OOD actions
>
> We thank the reviewer for this question. Due to the character limit, we refer the reviewer to our **response to Reviewer QNNU [w3]** for a detailed discussion of the robustness to OOD actions. The verifier shows **a certain degree of generalization to OOD actions**, predicts approximately half progress for half-speed actions and near zero mean value for random expert actions and corrupted actions.
>
> > **[w3]** The method lacks theoretical analysis and guarantees.
>
> We agree that the paper does not provide formal theoretical guarantees. Our contribution is primarily an empirical inference-time framework for improving pretrained robot policies.
>
> > **[q1]** Will the reconstruction error affect the overall accuracy?
>
> The Action-VAE in TapSampling is designed as an **efficient candidate generator** rather than a perfect reconstructor. Therefore, the reconstruction loss is not directly comparable to the final training loss of a VLA policy.
> Additionally, the subsequent action-verification step further filters out part of the low-quality candidates.
> While in principle a very large VAE reconstruction error could degrade the quality of sampled candidates, **moderate reconstruction error does not appear to significantly degrade final performance**. We conducted an analysis to empirically validate this.
>
> - We sample 4 policy actions and 12 VAE actions for each state, and divide the VAE actions into three groups (near, mid, and far) based on their nearest-neighbor distance to the policy samples, as an approximation to different levels of reconstruction-induced deviation. Each group contains 4 VAE actions. We then perform action scoring and final action generation using the 4 policy actions together with the 4 VAE actions from each group.
>
> *Due to the limited rebuttal timeline, we have not yet completed this additional analysis on the full 1000-task evaluation. We therefore report results on the first 200-task subset as a preliminary analysis. We are continuing the full evaluation. As the follow-up author response is triggered by additional reviewer comments, we will report the complete results if follow-up response is available.*
>
> |Method|1|2|3|4|5|Avg. Len|
> |-|-|-|-|-|-|-|
> |near|96.0|90.5|85.0|82.5|76.0|4.300|
> |mid|96.0|91.0|86.0|82.0|75.5|4.305|
> |far|96.0|89.5|83.5|83.5|78.5|4.310|
>
> Current results suggest that downstream performance remains highly similar across the three groups (Avg. Len 4.300/4.305/4.310), with **no clear degradation as the VAE samples move farther from the policy samples**. This suggests that, within the tested range, TapSampling is not sensitive to moderate candidate deviation from the policy samples.
>
> Taken together with the above analyses and the results in Tabs. 4 and 5, these findings suggest that downstream performance depends more on **distributional fidelity of the sampled actions** than on exact reconstruction accuracy. Moderate reconstruction error therefore does not appear to strongly affect final performance.
>
> > **[q3]** Ablation on selection threshold
>
> The performance is relatively stable across a reasonable range of thresholds.
> |Threshold|1|2|3|4|5|Avg. Len|
> |-|-|-|-|-|-|-|
> |0.04|96.8|92.9|89.0|85.3|80.1|4.441|
> |0.06|97.0|93.2|89.7|85.1|80.6|4.456|
> |0.08|96.5|92.9|89.4|86.4|81.1|4.463|
> |0.10|97.1|93.1|89.2|84.8|80.1|4.443|
>
> > **[q4]** Open-source plan
>
> We will release the training and inference code and the model checkpoints.

---

> > ### Author Rebuttal · Reviewer_gsdS · 2026-04-02
> >
> > Thank for your reply. I wil raise my score.

---

> > > ### Author Response · Authors · 2026-04-04
> > >
> > > We are pleased to see that our answers address your concerns, and thank you for your positive evaluation. Your insightful suggestions have helped us improve the quality of this paper.
> > >
> > > Additionally, we have completed the analysis of VAE reconstruction error for **[q1] Will the reconstruction error affect the overall accuracy?** on the full 1000-task evaluation. We would like to provide the results here. The experimental results show no clear performance degradation as the VAE samples move farther from the policy samples, which is consistent with the results observed during the first round of rebuttal.
> > >
> > > | Method | 1 | 2 | 3 | 4 | 5 | Avg. Len ↑ |
> > > |:---:|:---:|:---:|:---:|:---:|:---:|:---:|
> > > | near | 97.0 | 92.8 | 89.0 | 84.8 | 79.8 | 4.434 |
> > > | mid | 96.9 | 93.2 | 88.8 | 84.5 | 79.8 | 4.432 |
> > > | far | 96.8 | 92.8 | 88.4 | 85.3 | 80.1 | 4.434 |

---

### Official Review · Reviewer_QNNU · 2026-03-12

**Soundness:** 3
**Presentation:** 3
**Significance:** 2
**Originality:** 2
**Overall Recommendation:** 4
**Confidence:** 3

**Summary:**

This paper proposes TapSampling, a plug-and-play inference-time sampling framework for improving generalist robotic manipulation policies without fine-tuning. The framework consists of two components: an Action-VAE that encodes policy-generated actions into a low-dimensional latent space; and (2) a task-progress-understanding verifier trained on intrinsic sequential information from expert trajectories to predict how candidate actions affect task completion progress. Experiments on CALVIN, LIBERO-Long, and real-world manipulation tasks demonstrate consistent improvements across multiple policy architectures.

**Compliance With Llm Reviewing Policy:**

Affirmed.

**Final Justification:**

All my concerns are well resolved. I decided to maintain my score to weak accept.

**Key Questions For Authors:**

see the weaknesses

**Limitations:**

yes

**Strengths And Weaknesses:**

**Strengths:**

1) The framework is policy-agnostic and requires no fine-tuning of the base policy, which makes it practically useful. The authors demonstrate compatibility with four distinct policy architectures, spanning diffusion-based, autoregressive, and flow-matching paradigms, which strengthens the generality claim.

2) The task-progress formulation for the verifier is well-motivated. By leveraging the intrinsic sequential structure of expert trajectories to construct positive and negative training samples, the method avoids the need for additional manual annotation or expensive data synthesis. The predicted scores carry interpretable semantic meaning, which is a notable advantage over prior verifiers that produce opaque scores.

**Weaknesses:**

1) The magnitude of improvement, while consistent, is relatively modest in several settings. For instance, the improvement on VPP and on the first task of CALVIN across all policies is small. Appendix A shows that the retry-based upper bound for VPP is around 4.72, suggesting TapSampling captures only a fraction of the potential headroom. The paper would benefit from a deeper discussion of why the gap remains large and under what conditions the method is most or least effective.

2) The linear task-progress assumption is a strong simplification. Many manipulation tasks involve phases of varying difficulty or progress rates (e.g., approaching an object may constitute minimal progress while the actual grasp is a critical moment). The paper does not discuss how sensitive the verifier is to violations of this assumption, nor does it explore alternative progress models.

3) The negative sample construction via action reversal is intuitive but may not always produce semantically meaningful negative examples. Reversing end-effector pose deltas is geometrically reasonable, but for actions involving discrete gripper state changes or contact-rich manipulation, simple reversal may not faithfully represent "bad" actions. There is no analysis of the quality or diversity of the constructed negatives.

---

> ### Author Rebuttal · Authors · 2026-03-31
>
> We thank the reviewer for the thorough comments. We respond to the concerns about improvement, linear task-progress assumption, and OOD problem point by point.
>
> > **[w1]** Modest improvement and remaining gap to retry upper bound
>
> **1.1 Modest gains in some settings** (e.g. CALVIN VPP 1st task (96.4%→96.5%)). In some settings, especially the first task in CALVIN, the base policy already achieves a high success rate, leaving limited room for absolute improvement. By contrast, the benefit becomes more pronounced on **harder and longer-horizon tasks**, where local action errors accumulate over time. For example, the improvements are substantially larger on the task 5 of CALVIN (Tab. 1, +4.2%/+6.4%/+2.8%).
>
> **1.2 Remaining gap to the Retry Upper Bound.** We agree that Appendix A reveals substantial remaining headroom. However, we believe this gap should be interpreted in light of the fundamentally different evaluation protocols. The Retry Upper Bound is better viewed as a **trajectory-level pass@k result**: it allows multiple independent full-trajectory attempts, relies on oracle success/failure signals from the simulator, and resets the environment to a clean initial state after each failure. In contrast, TapSampling is designed to **improve pass@1** within a single continuous rollout. This distinction is especially important for real-world deployment, where oracle reset is typically unavailable. This highlights the core motivation of our work: trading inference-time computation for a higher pass@1 success rate.
>
> **1.3 When is TapSampling most or least effective?** TapSampling is most effective on long-horizon tasks where local action errors are more likely to accumulate and affect downstream success. In contrast, its gain is naturally smaller on easier or near-saturated tasks.
>
>
> > **[w2]** Linear task-progress assumption and sensitivity to violations
>
> Our intent is not to claim that true task progress evolves uniformly over time, but to use normalized trajectory time as an approximation of progress, serving as a continuous and annotation-free supervision proxy for training the action verifier.
>
> For inference-time sampling, the key requirement is not a perfectly calibrated global progress model, but a **useful same-state action-scoring signal** that helps distinguish more promising actions from poorer ones. Since the verifier scores short-horizon candidate actions at inference time, moderate mismatch between true progress and the linear target may affect the calibration of the predicted score, while still preserving a useful local preference signal for action scoring.
>
> **What does the verifier learn when the linear assumption is violated?** Our method relies on the temporal order of successful expert demonstrations to provide supervision for action scoring. As a result, even when true progress is not perfectly linear across task phases, the verifier can still learn **which kinds of actions successful expert trajectories tend to use to complete the task**. In such cases, the predicted score should be understood as a noisy demonstration-derived progress proxy rather than an exact measure of physical task progress. In particular, our experiments suggest that this simple proxy is sufficient to provide a useful local ranking signal for action selection.
>
> > **[w3]** Performance on OOD actions
>
> Reversed actions do not cover all semantically invalid actions. However, our goal is not to exhaustively model all failure modes, but to provide a contrastive signal that separates expert-like actions from poor candidates.
>
> We evaluated the verifier (trained on CALVIN ABC) on environment D by scoring five types of actions to assess its robustness: (1) forward actions from expert data, (2) reversed actions (backward), (3) half-speed actions, (4) random action chunks from expert data, and (5) gripper-corrupted actions. Forward and reversed actions receive the highest and lowest average scores, respectively. **Half-speed actions score roughly half as much as forward actions. Random and corrupted have average scores close to zero**. Approximately 80% of bad actions are filtered with a threshold of 0.08.
> **The verifier shows a certain degree of generalization to OOD actions.**
>
> |Action|Mean|-0.20~-0.16|-0.16~-0.12|-0.12~-0.08|-0.08~-0.04|-0.04~0.00|0.00~0.04|0.04~0.08|0.08~0.12|0.12~0.16|0.16~0.20|
> |-|-|-|-|-|-|-|-|-|-|-|-|
> |forward|0.0771|0.2%|1.3%|2.4%|4.5%|8.2%|11.0%|13.1%|26.1%|24.7%|7.9%|
> |reversed|-0.0713|8.1%|26.4%|22.9%|12.5%|9.3%|7.1%|5.7%|4.1%|2.5%|0.7%|
> |half_speed|**0.0477**|0.1%|1.2%|2.9%|6.2%|8.7%|14.7%|34.2%|27.0%|4.5%|0.5%|
> |random|**-0.0020**|0.9%|7.4%|16.0%|15.8%|12.4%|12.8%|12.2%|11.0%|9.9%|1.7%|
> |corrupted|**-0.0198**|7.7%|11.9%|11.8%|10.7%|10.1%|12.1%|11.6%|11.2%|7.2%|2.8%|
>
> We further computed the average ranking of the five action types, which shows: **forward > half-speed > random > corrupted > reversed**.
> |forward|reversed|half_speed|random|corrupted|
> |-|-|-|-|-|
> |1.814|4.121|2.588|3.071|3.406|

---

> > ### Author Rebuttal · Reviewer_QNNU · 2026-04-01
> >
> > All my concerns are well resolved. I decided to maintain my score to weak accept.

---

> > > ### Author Response · Authors · 2026-04-04
> > >
> > > Thanks for your feedback. We are pleased that our response has helped address your concerns.

---

### Official Review · Reviewer_ufei · 2026-03-12

**Soundness:** 2
**Presentation:** 3
**Significance:** 2
**Originality:** 2
**Overall Recommendation:** 4
**Confidence:** 4

**Summary:**

This paper proposes TapSampling, a plug-and-play inference-time framework for improving non-deterministic robotic manipulation policies without fine-tuning the base policy. The method has two components: (1) an Action-VAE that builds a low-dimensional learned posterior from a small number of policy-sampled actions and decodes additional candidate actions efficiently, and (2) a task-progress verifier that scores candidate actions by predicting expected progress toward task completion. The verifier is trained from the intrinsic temporal structure of expert trajectories, using forward subsequences as positive examples and reversed subsequences as negative examples.

**Compliance With Llm Reviewing Policy:**

Affirmed.

**Final Justification:**

The rebuttal has largely addressed my previous concern, and I am happy to raise my rating to weak accept.

**Key Questions For Authors:**

The verifier target is derived from the assumption that task progress increases linearly with trajectory time. How robust is the method when this assumption is violated, such as in tasks with long setup phases, phase transitions, or temporary regressions?

**Limitations:**

Yes

**Strengths And Weaknesses:**

Strengths
- Intuitive test-time method: the sampling and verification decomposition is easy to understand, and the use of task-progress prediction gives the verifier a more interpretable semantic target than a scalar reward. The overall method is also reasonably policy-agnostic.
- Strong empirical result: the experiments cover several policy classes (Diffusion Policy, OpenVLA, VPP, and π0.5), two simulation benchmarks, and real-world robot tasks. The gains are generally consistent, and the paper also includes latency comparisons, sampling ablations, and verifier analyses.

Weakness
- The verifier relies on a strong linear-progress assumption. The training target assumes that task progress increases linearly along expert trajectories. This is a convenient source of supervision, but it may be a poor approximation for many manipulation tasks with long preparation phases, contact-rich transitions, or multi-stage structure. In such cases, temporal position may be only a weak proxy for true progress, which could inject label noise into verifier training. The paper would be stronger with either a discussion of this limitation or an analysis on tasks where progress is clearly non-linear or partially non-monotonic.
- The paper is insufficiently positioned with respect to recent literature. There has been a growing body of work on guided test-time sampling [1-3], latent space steering [4], and progress reward from reverse sequence [5]. It is not fully clear which aspects of TapSampling are genuinely novel. A more careful discussion of the connections and distinctions would help clarify its originality.

[1] Bidirectional Decoding: Bidirectional Decoding: Improving Action Chunking via Guided Test-Time Sampling, ICLR 25 \
[2] Strengthening Generative Robot Policies through Predictive World Modeling, RAL 25 \
[3] DynaGuide: Steering Diffusion Polices withActive Dynamic Guidance, NeurIPS 25 \
[4] Steering Your Diffusion Policy with Latent Space Reinforcement Learning \
[5] ReWiND: Language-Guided Rewards Teach Robot Policies without New Demonstrations

---

> ### Author Rebuttal · Authors · 2026-03-31
>
> We thank the reviewer for the careful reading and constructive feedback. We respond to the concerns on the linear-progress assumption and related-work positioning below.
>
> > **[w1q1]** Robustness when the linear-progress assumption is violated
>
> In the extreme case where the linear-progress assumption is severely violated, we expect the main impact to be on interpretability rather than effectiveness: the predicted score may no longer faithfully reflect true task progress. In this case, the verifier behaves more like a demonstration-induced action scorer, **capturing which kinds of actions successful expert trajectories tend to act** to complete the task.
>
> We agree that true task progress for some tasks (e.g. tasks with long preparation phases) is not exact linear. However, for inference-time sampling, the key requirement is not a perfectly calibrated global progress model, but a useful same-state action-scoring signal that helps distinguish more promising actions from poorer ones. In our method, the normalized-time target is intended as a practical continuous self-supervision signal for the action-conditioned verifier, rather than a claim that physical task progress is strictly linear. More precisely, the verifier is trained with a demonstration-induced proxy derived from the temporal order of successful expert trajectories. We believe this proxy can still provide a useful signal for our inference-time action-scoring objective for two reasons.
>
> - Our verifier is designed for same-state candidate scoring at inference time. Its outputs are used for **local action evaluation**, rather than as a **globally calibrated progress estimator** across all task phases. In this setting, what matters most is that the verifier provides a useful local signal to separate poor candidates from promising ones under the same environment state.
> - The verifier scores short-horizon action chunks rather than full trajectories. Therefore, it does not require a perfectly calibrated global progress measure. A local signal is often sufficient to distinguish actions that move the task forward from those that hinder it. In particular, our supervision is constructed from expert demonstrations, so even when true progress is not linear, the labels still reflect **which kinds of actions successful expert trajectories tend to act** to complete task.
>
> **Revision.** We will clarify this limitation in the revised paper and add further discussion and analysis on the linear-progress assumption. We will also note that related embodied RL/value-learning works, such as VLAC[1] and pi0.6[2], similarly use normalized-time or steps-to-go style supervision, although in a state-scoring setting for reinforcement learning rather than an action-scoring setting.
>
> [1] A Vision-Language-Action-Critic Model for Robotic Real-World Reinforcement Learning
>
> [2] $\pi_{0.6}^{*}$ a VLA That Learns From Experience
>
> > **[w2]**: Difference with respect to recent literature
>
> Inference-time sampling shares certain similarities with frozen-policy steering and progress-reward RL. We will revise the related work section to better clarify how TapSampling differs from these approaches and thereby address the confusion.
>
> TapSampling differs from these adjacent methods in several key ways.
>
> (1) BID uses **hand-crafted guided** decoding criteria to select action chunks at test time, whereas TapSampling **learns an action-conditioned verifier** for candidate action scoring.
>
> (2) GPC **performs future rollout and evaluates** candidate actions through a predictive world model and future-outcome scoring, which makes inference relatively slow, whereas TapSampling **directly scores action chunks** without rollout.
>
> (3) DynaGuide applies guidance **inside the diffusion denoising process**, rather than performing explicit multi-sampling and candidate scoring, which limits it to diffusion-based policies.
>
> (4) DSRL learns a **noise proposal module via reinforcement learning** to steer a frozen diffusion policy throught generating better initial noise, whereas TapSampling performs inference-time sampling and selection without learning a policy-specific proposal module.
>
> (5) ReWiND learns a **video-sequence reward model** for RL-based policy optimization rather than direct action evaluation or inference-time sampling, whereas TapSampling learns an **action-conditioned scorer** that enables inference-time selection over candidate actions.

---

> > ### Author Rebuttal · Reviewer_ufei · 2026-04-04
> >
> > Thank you for the rebuttal. It has largely addressed my concern, and I am happy to raise my score by 1.

---

> > > ### Author Response · Authors · 2026-04-04
> > >
> > > Thanks for your feedback. We are delighted to know that our responses have helped clarify and address your concerns.

---

### Official Review · Reviewer_PG8f · 2026-03-16

**Soundness:** 3
**Presentation:** 3
**Significance:** 2
**Originality:** 2
**Overall Recommendation:** 4
**Confidence:** 4

**Summary:**

This paper presents the TapSampling framework, which adds 2 new capabilities to pretrained generalist robot policies: 1) efficient action sampling conditioned on the current state compared to naively drawing more samples from the policy distribution, and 2) a scoring model that allows the evaluation and selection among sampled action chunks.

First, for efficient action sampling, the authors train a VAE with Gaussian posteriors for chunked actions. This allows them to draw many more sample action trajectories from only a few actions from the original policy. The motivation of this is the observation shared in Figure 4, where the sampling latency grows as the sample size increases. The VAE allows the authors to sample from a mixture of Gaussian posteriors and then decode them back quickly.

Next, the progress predictor is trained on both readily available positive examples, which are the human demonstrated actions, and smartly defined negative actions. Because actions in robot manipulation are usually EEF pose targets, the authors can compute the negation of them and put them in reverse order to provide negative supervision. The progress predictor can then be used to select the best action chunk from the sampled ones.

The TapSampling method is evaluated on simulated and real robot tasks. The main results show that when combined with generalist models, trained or finetuned on these tasks, this sampling, scoring and selection steps can introduce some additional performance gain.

**Compliance With Llm Reviewing Policy:**

Affirmed.

**Final Justification:**

The OOD action results provided in the rebuttal are convincing: although only trained on forward/backward action, the critic effectively scores other actions.

Other clarifications and explanations also addressed my concern on inference latency and improvements over naive baseline.

Hence, I have raised my score by 1.

**Key Questions For Authors:**

- Minor issue: paper title is still the template title.
- How does the verifier's performance compare to an offline Q learning critic?
- The latency growth trend is a bit hard to see in Figure 4 because the x, y coordinates grow at different scale. Also, doesn't VAE decoding cause additional latency? For sampling the original model, it is also possible to just scale the hardware to achieve somewhat constant latency right?

**Limitations:**

The authors could elaborate more on the limitations. For example, the final policy performance is still largely constrained by the original generalist model. Also, because of the VLA backbone, the verifier likely require significant compute for training.

**Strengths And Weaknesses:**

Strengths:
1. The main TapSampling framework is well motivated and explained clearly.
2. The algorithm is not tied to a specific generalist policy. As long as the policy predicts action chunks, this is compatible.
3. The algorithm constantly improves upon the out-of-the-box performance of various models.

Weaknesses:
1. The main novelty of this work seems to only be the negative action sequence generation trick used to train the progress predictor. Action Chunking Transformers (ACT) has used VAEs for action latent spaces. And the idea of learning the q value function for action candidate selection is also not new.
2. The gain of the TapSampling framework seems marginal. As reported in Table 4, the learned posterior sampling performance is very similar to gaussian sampling.
3. The verifier model is trained only on +- k/t data, it is unclear how well it performs outside this training domain. For example, what if an action chunk just does the task at half the speed? Does the model predict half the progress?

---

> ### Author Rebuttal · Authors · 2026-03-31
>
> We thank the reviewer for the insightful comments and helpful suggestions. We respond to the concerns below.
>
> > **[w1]** Positioning relative to ACT and prior action verifiers
>
> The main novelty of TapSampling lies at the framework level: it formulates an inference-time sampling framework where **both components (Action-VAE and Action-Scorer) play roles distinct from prior work**. We'll revise the related works section to make this distinction from VAE-based policies and prior action verifiers more explicit.
>
> **1.1 Difference from ACT.** Compared with other methods with VAE latent space, Action-VAE is not used as a module inside the policy for state-to-action learning. Instead, it is a policy-sample-conditioned proposal model. A small number of policy actions are encoded into action-dependent posteriors to form a latent distribution for candidate expansion. Specifically, ACT [1] uses a static prior $z=0$ for state-to-action map, while TapSampling uses action-depended posterior for action-to-action sampling.
>
> **1.2 Difference from Q/reward models.** Our verifier is not a standard state-conditioned reward/Q model for policy training but an action-conditioned scorer trained directly for inference-time selection. The training signal is constructed from the sequential structure of demonstration trajectories without rewards, preference labels, or offline RL. This yields semantically meaningful scores, enabling plug-and-play inference-time evaluation across different policies.
>
> Therefore, the contribution is not the isolated use of VAE or value estimation, but their new formulation and interaction for plug-and-play inference-time improvement of pretrained policies.
>
>
> > **[w2]** Marginal gain. In Table 4, the learned posterior sampling performance is very similar to gaussian sampling.
>
> **Table 4 does not measure the gain of the full TapSampling framework**. Instead, it isolates the effect of the action-sampling strategy within TapSampling, since both Gaussian and Learned Posterior Sampling use the same TapSampling verifier. The full framework gain should be measured against the base policy (VPP 4.39->4.46), while 4.43->4.46 reflects the benefit of learned posterior sampling over a Gaussian alternative within the same framework.
>
> Even for the sampler comparison, we believe the gain is meaningful. The learned posterior achieves this improvement with almost the same inference cost as Gaussian sampling (0.488s vs. 0.465s), while yielding clearer benefits on the long-horizon subtasks (+1.6%/1.2% on fourth/fifth tasks). Additionally, even core designs in strong models on CALVIN often yield modest absolute gains in Avg. Len. (e.g. [2] segment cues 4.40→4.44). Therefore, we believe the gain from learned posterior sampling is meaningful.
>
> > **[w3]** Performance on OOD actions
>
> We thank the reviewer for this question. Due to the character limit, we refer the reviewer to our response to **Reviewer QNNU [w3]** for a detailed discussion of the robustness to OOD actions. The verifier shows a certain degree of generalization to OOD actions and **predicts approximately half progress for half-speed actions** (0.077 vs. 0.048 mean scores for forward and half-speed actions).
>
>
> > **[q1]** Paper title issue
>
> Thanks. We'll correct this in the revised version.
>
> > **[q2]** Comparison with offline Q learning critic
>
> We train a Q critic using the same architecture as our verifier for a fair comparison. Key configurations (e.g. reward definition) follow V-GPS[3].
> Our verifier performs slightly better than the Q critic, with a higher Avg. Len. and better success rates for the second to fifth tasks.
> |Method|1|2|3|4|5|Avg. Len|
> |-|-|-|-|-|-|-|
> |Q-critic|**96.9**|92.5|89.1|85.2|80.3|4.440|
> |Ours|96.5|**92.9**|**89.4**|**86.4**|**81.1**|**4.463**|
>
> > **[q3]** VAE latency and hardware scaling
>
> We will optimize the presentation of Fig.4. The lightweight Action-VAE (6.6M params) introduces minimal latency overhead (0.319→0.328s), as shown in Fig. 4.
> We agree that scaling hardware is a viable strategy. Tab. 4 shows that drawing more samples from the policy further improves the performance of TapSampling (4.46→4.50).
> However, scaling hardware is often constrained in practice. The lightweight Action-VAE provides an efficient way to expand candidates with minimal latency overhead.
>
> > **[Limit]** Discussion of limitations
>
> We thank the reviewer for the suggestion and will add a more detailed discussion of limitations. TapSampling is designed to enhance pretrained policies via better action selection, and thus remains bounded by the underlying policy performance. Regarding compute, training the verifier takes approximately 10 hours on 4×A800 GPUs.
>
> [1] Learning Fine-Grained Bimanual Manipulation with Low-Cost Hardware.
>
> [2] DreamVLA: A Vision-Language-Action Model Dreamed with Comprehensive World Knowledge.
>
> [3] Steering Your Generalists: Improving Robotic Foundation Models via Value Guidance.

---

> > ### Author Rebuttal · Reviewer_PG8f · 2026-04-04
> >
> > Thanks for the detailed response. I appreciate the extra study on the OOD actions: the half-speed data directly answered my question.
> >
> > Thanks also for addressing my other points, specifically:
> > - Clarifying the framework's novelty compared to ACT and standard Q-models.
> > - Explaining the Table 4 numbers and showing the VAE's minimal latency overhead.
> > - Running the new baseline comparison against the offline Q-critic.
> >
> > I'll raise my score.

---

> > > ### Author Response · Authors · 2026-04-04
> > >
> > > We are glad that our response helped clarify your concerns. Your valuable comments have helped us improve the manuscript.

---

### Decision · Program_Chairs · 2026-04-30

**Decision:**

Accept (regular)

**Comment:**

This paper introduces TapSampling, a framework that enhances the performance of pretrained robotic manipulation policies during inference. It consists of an Action-VAE for efficient candidate action sampling and a task-progress verifier that scores these candidates based on predicted completion progress. Key strengths include its policy-agnostic nature, interpretable scoring, and consistent performance gains across diverse architectures and environments. However, weaknesses noted include the modest magnitude of improvement in some settings and the reliance on a simplified linear-progress assumption for verifier training.

The authors addressed several technical concerns, including the novelty of their framework compared to existing VAE-based policies and standard value models. They clarified that their Action-VAE acts as a proposal model for candidate expansion rather than a core policy component. Regarding the linear-progress assumption, they argued that while it is an approximation, it provides a sufficient local ranking signal for effective action selection at inference time. They also provided additional experimental results on out-of-distribution actions and a comparison against an offline Q-learning critic to demonstrate robustness.

I recommend accepting this paper, as the authors successfully addressed the initial reviewer's concerns about novelty, latency, and the robustness of the progress-based verifier. The framework demonstrates a practical, effective way to improve existing robotic models without costly fine-tuning.